# Chromosome-level genome assembly and population genomics of *Robinia pseudoacacia* reveal the genetic basis for its wide cultivation

Zefu Wang[1,2,3,5], Xiao Zhang [2,5✉], Weixiao Lei[1], Hui Zhu [1], Shengdan Wu [1✉], Bingbing Liu [4✉] & Dafu Ru [1✉]

Urban greening provides important ecosystem services and ideal places for urban recreation and is a serious consideration for municipal decision-makers. Among the tree species cultivated in urban green spaces, *Robinia pseudoacacia* stands out due to its attractive flowers, fragrances, high trunks, wide adaptability, and essential ecosystem services. However, the genomic basis and consequences of its wide-planting in urban green spaces remains unknown. Here, we report the chromosome-level genome assembly of *R. pseudoacacia*, revealing a genome size of 682.4 Mb and 33,187 protein-coding genes. More than 99.3% of the assembly is anchored to 11 chromosomes with an N50 of 59.9 Mb. Comparative genomic analyses among 17 species reveal that gene families related to traits favoured by urbanites, such as wood formation, biosynthesis, and drought tolerance, are notably expanded in *R. pseudoacacia*. Our population genomic analyses further recover 11 genes that are under recent selection. Ultimately, these genes play important roles in the biological processes related to flower development, water retention, and immunization. Altogether, our results reveal the evolutionary forces that shape *R. pseudoacacia* cultivated for urban greening. These findings also present a valuable foundation for the future development of agronomic traits and molecular breeding strategies for *R. pseudoacacia*.

[1] State Key Laboratory of Herbage Improvement and Grassland Agro-Ecosystem, College of Ecology, Lanzhou University, Lanzhou 730000, China. [2] Tianjin Key Laboratory of Conservation and Utilization of Animal Diversity, College of Life Sciences, Tianjin Normal University, Tianjin 300387, China. [3] Co-Innovation Center for Sustainable Forestry in Southern China, College of Biology and the Environment, Nanjing Forestry University, Nanjing 210037, China. [4] Institute of Loess Plateau, Shanxi University, Taiyuan 030006, China. [5] These authors contributed equally: Zefu Wang, Xiao Zhang. ✉email: zhangxiao@tjnu.edu.cn; wusd@lzu.edu.cn; lbb2015@sxu.edu.cn; rudf@lzu.edu.cn

Urban greening refers to the organized or semi-organized construction of green infrastructures like urban green spaces, street trees, and hedges in cities that provide the ideal environment for urban recreation and acts as an important ecosystem service to control pollution, regulate temperature, and manage stormwater[1–3]. With the acceleration of urbanization in low-income and lower-middle-income countries, urban greening is attracting more attention from municipal decision-makers and landscape planners[4,5].

In China, $2.5 \times 10^4$ km$^2$ of built-up area (BUA) increased from 2001 to 2018 and corresponds to 47.5% of the global increase and represents the fastest speed of urbanization in the world[6]. The demand for green infrastructure, particularly in terms of urban planting, has become increasingly important for Chinese city designers due to rapid urbanization and the desire to meet the recreational needs and landscape perceptions of urbanites[3]. Their achievements marked China as the greatest contributor to urban greening for land coverage in the world from 2001–2018[6]. Recently, other studies have shown that urban environmental change can influence four evolutionary processes: mutation, genetic drift, gene flow, and adaptation due to natural selection[7,8]. These urban areas represent novel ecosystems where green infrastructure construction provides precious opportunities for researchers to observe how genetic and phenotypic effects can accompany the urban greening process introduced by human activity during urbanization[9].

Over the last few decades, *Robinia pseudoacacia* (black locust) has become one of the most widely cultivated and popular woody species in urban green areas in China[10–12]. These outcrossing, fast-growing, and nitrogen-fixing legume trees belong to the Faboideae subfamily, Fabaceae family, and originated in North America. They were then introduced to sub-Mediterranean and temperate regions, including continental Europe, Australia, and East Asia (of which China alone possesses over one million ha of plantation)[13,14]. In 2010, the estimated area of *R. pseudoacacia* plantations outside their native range was about 3 million ha, and the number keeps growing[15]. Currently, the species has become the second most widely planted broad-leaved tree species in the world, following Eucalyptus spp[16].

*R. pseudoacacia* was first introduced to China and planted in Nanjing and Qingdao during the late 19th century[17]. Compared to other plants in urban green spaces (such as *Trifolium repens*, *Cinnamomum camphora*, and *Ligustrum lucidum*), *R. pseudoacacia* is favored by Chinese urbanites because of their rich flowers, attractive fragrances, deciduous broad leaves, and high trunks[18]. Additionally, its high tolerance to a wide range of soil conditions[19], and high adaptability to harsh environments and low fertility enable it to thrive in urban areas across vast climatic regions while providing essential ecosystem services[11,12,20–22]. For example, in the ecologically fragile Loess Plateau, human-planted *R. pseudoacacia* covers >70,000 ha[23,24], and has been shown to notably alter vegetation structures, soil properties, and microbial biomass and activities[19,25]. In Shenyang, one of the largest cities in northeastern China, *R. pseudoacacia* has also played an important role in improving air quality by absorbing ambient fine particulate matter with a diameter of ≤2.5 μm (PM2.5), which acts as a primary air pollutant that causes human disease[12].

While the environmental effects and morphology of *R. pseudoacacia* in urban ecosystems have been well-researched in the past, we currently do not know much about the genomic basis and consequences of its wide distribution in urban green spaces. Notably, it is essential that we identify an accurate, complete, and contiguous genome assembly for this species, which to understand its valuable genetic variation, and apply cutting-edge molecular biology technologies. These limitations further obstruct in-depth molecular breeding of *R. pseudoacacia* and the proper introduction of *R. pseudoacacia* during green infrastructure construction in cities.

Here, we created a chromosome-level genome assembly of *R. pseudoacacia* that was deciphered using integrated Illumina short-read sequencing, Nanopore long-read sequencing, and chromosomal conformational capture (Hi-C) technologies. We characterized its genome in detail, including genomic structure, gene annotation, and repeat sequences. We also conducted a whole-genome re-sequencing analysis of 59 *R. pseudoacacia* individuals across 14 Chinese cities. Our following comprehensive population genomic survey revealed genetic relationships among all the individuals, and inferred their demographic histories. Additionally, we identified selection signatures that may be involved in urban planting. Together, our study provides a valuable resource to facilitate comparative genomics, adaptive evolution studies, and genomic-assisted breeding for *R. pseudoacacia*, and improves our general understanding of urban planting.

## Results and discussion

**Genome assembly and annotation**. To generate a chromosome-level genome for *R. pseudoacacia* (Fig. 1a), the genome of a single *R. pseudoacacia* individual sampled from Lanzhou, China (36°2′57″N, 103°51′34″E) was sequenced using a hybrid strategy that combined Oxford Nanopore long-read sequencing, Illumina paired-end sequencing, and Hi-C technologies. Specifically, we obtained 75.6 Gb of long-read data with a read N50 of 31.7 kb, which confirms the high quality of our Oxford Nanopore sequencing libraries (Supplementary Table 1). Based on this data, we first assembled a contig-level genome assembly of 682.4 Mb, and then polished it using high-accurate Illumina reads. This assembly contained only 55 contigs with a contig N50 of 32.1 Mb, which indicates its high continuity (Table 1 and Supplementary Table 2). Our K-mer analysis based on the 111.34 Gb of Illumina paired-end data estimated that the genome was approximately 693.1 Mb with a heterozygosity rate of 1.13% (Supplementary Fig. 1 and Supplementary Table 3) that is consistent with our contig-level assembly.

To further improve the genome assembly, we established a Hi-C library and obtained 78.27 Gb of Hi-C reads. The assembly of *R. pseudoacacia* was successfully anchored onto 11 chromosomes with an improved N50 of 59.9 Mb that covered 99.3% of the raw assembly (Fig. 1b, Supplementary Fig. 2, and Supplementary Table 4), which suggests that most regions of the *R. pseudoacacia* genome were successfully assembled at the chromosome level. Our final assembly showed better continuity than most of the other genomes of the Fabaceae species (Supplementary Table 5).

The high quality of this genome was confirmed by multiple methods. First, we performed BUSCO analyses to determine the completeness of this genome assembly, which reported a complete score of 98.20% and indicates high completeness (Supplementary Table 6) and low redundancy of haplotype sequences (Supplementary Table 7). We also aligned Illumina paired-end whole-genome re-sequencing data obtained from other individuals and transcriptome data to this assembly. More than 97.9% of the re-sequencing data and more than 96.5% of the transcriptome data were successfully mapped to the assembly, which suggests high accuracy for this assembly.

We annotated repetitive sequences in *R. pseudoacacia* genome by combining both ab initio and homology-based methods. Over 405.9 Mb of sequences were identified as repetitive elements. Together, they constituted 59.47% of the genome assembly that was predominately LTRs (Supplementary Table 8). These results are consistent with previous observations in closely-related Legume species *Lotus japonicus*[26], in which LTRs are also the most abundant type of repeat elements. Intriguingly, although the genome of the *R. pseudoacacia* had a larger size than the closely-related *Lupinus albus* and *Cicer arietinum*, the proportion

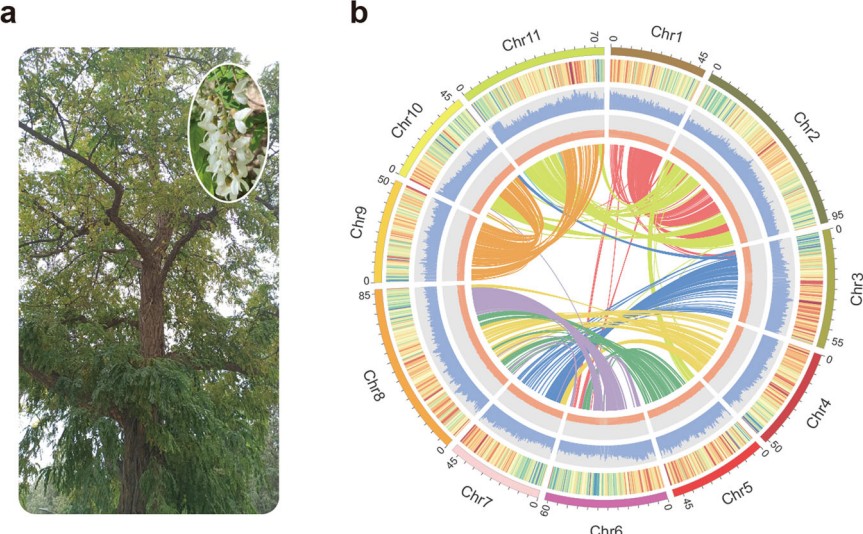

**Fig. 1 Sampling and genome assembly of *R. pseudoacacia*. a** The sequenced individual of *R. pseudoacacia* from Lanzhou University, Gansu Province, China (36°2′57″N, 103°51′34″E). The flowers of *R. pseudoacacia* were demonstrated in the figure. **b** Genome features from the *R. pseudoacacia* assembly. From outer to inner: (1) genome chromosomes, (2) gene density, (3) repeat density, (4) GC (guanine-cytosine) content, and (5) synteny information.

**Table 1 Assembly and annotation features from the *R. pseudoacacia* assembly.**

| Type | Statistics |
| --- | --- |
| Assembly size (bp) | 682,400,408 |
| Number of scaffolds | 20 |
| Scaffold N50 size (bp) | 59,867,354 |
| Number of contigs | 55 |
| Contig N50 size (bp) | 32,134,000 |
| Number of chromosomes | 11 |
| Ordered and oriented genome size (bp) | 677,388,330 |
| and percentage (%) | 99.27 |
| Repeat region size (bp) | 405,857,565 |
| and percentage (%) | 59.47 |
| GC content (%) | 33.33 |
| Number of protein-coding genes | 33,187 |
| Functional annotated genes (%) | 100.00 |

(59.47%) of repetitive elements was still similar to what was measured in these two species. Specifically, *Lupinus albus* has a 451 Mb genome that contains 60% repetitive elements[27], and *C. arietinum* has a 545 Mb genome that contains 60% repetitive elements[28]. The primary reason for this difference in genome size can be attributed to the influence of long terminal repeat retrotransposons (LTR-RTs) on genome size and evolution[29] (Supplementary Fig. 3). LTR-RTs are responsible for over 75% of the genome in some plant species and have been identified as a major driving force behind genome expansions[30]. These results suggest that the large size of the *R. pseudoacacia* genome may be partially driven by the explosion of repetitive elements (Supplementary Fig. 3).

Using a customized gene prediction pipeline that incorporated ab initio, homology, and transcriptome-based approaches, we predicted 33,187 protein-coding genes in the *R. pseudoacacia* genome (Supplementary Table 9). The average length of the genes was 4492 bp with an average of 5 exons (Supplementary Table 9). We matched these predicted genes to functional annotations deposited in five databases, including TrEMBL, SWISS-PROT, Gene Ontology (GO), Kyoto Encyclopedia of Genes and Genomes (KEGG), and InterPro. More than 99% of these genes were functionally annotated (Supplementary Table 10). Our official gene set for *R. pseudoacacia* genome covered more than 97% of BUSCO core genes (Supplementary Table 11), which suggests that this gene set was robust, and that most of the genes in this genome were functionally conserved. We further investigated the syntenic genes based on this gene set and found that there were 2357 syntenic blocks that ranged in size from 5 to 393 gene pairs in the *R. pseudoacacia* genome (Fig. 1b), which indicates the occurrence of whole-genome duplication (WGD) events.

**Phylogenetics and genome evolution**. To reveal the phylogenetics of *R. pseudoacacia* and its evolutionary trajectories, we compared its genome with those of 14 plant species of the family Fabaceae and used *Arabidopsis thaliana* as an outgroup. Based on 273 strictly single-copy orthologous genes from these 16 plant genomes, we established a genome-scale phylogenetic tree using maximum likelihood (ML) methods (Fig. 2a). Our phylogenetic tree showed that *R. pseudoacacia* was located close to the clade consisting of *Medicago truncatula*, *Trifolium pratense*, *C. arietinum*, and *L. japonicus*. These results are in accordance with other previous studies[31,32]. By estimating the divergence time for each node based on the 4DTv sites from the single-copy orthologous genes, we found that the divergence time between the *R. pseudoacacia* and other four species (*M. truncatula*, *T. pratense*, *C. arietinum*, and *L. japonicus*) was about 41.31 million years ago (Ma).

To provide insights into the evolution history of the *R. pseudoacacia* genome, we compared its synonymous substitution rates (Ks) with genomes of *G. max* and *M. truncatula* to identify the homologous gene pairs and to detect syntenic relationships between these species. The rate of Ks curves of collinear gene pairs suggested that the genomes of *R. pseudoacacia* underwent two rounds of whole-genome duplication (WGD). The older event, known as the γ event, is shared by all core eudicot lineages, while the more recent event occurred ~59 million years ago and is shared by all species in the Faboideae subfamily[33] (Fig. 2b). Notably, unlike *G. max*[34], *R. pseudoacacia* did not experience a species-specific WGD event (Fig. 2b). Furthermore, we detected a total of 88,836 gene pairs among these three species, and classified them into 2304 syntenic blocks, covering 662 Mb (97.73%) of the *R. pseudoacacia* assembly. Consistent with the Ks analysis, we also found a 2:4 syntenic relationship between *R. pseudoacacia* and *G.*

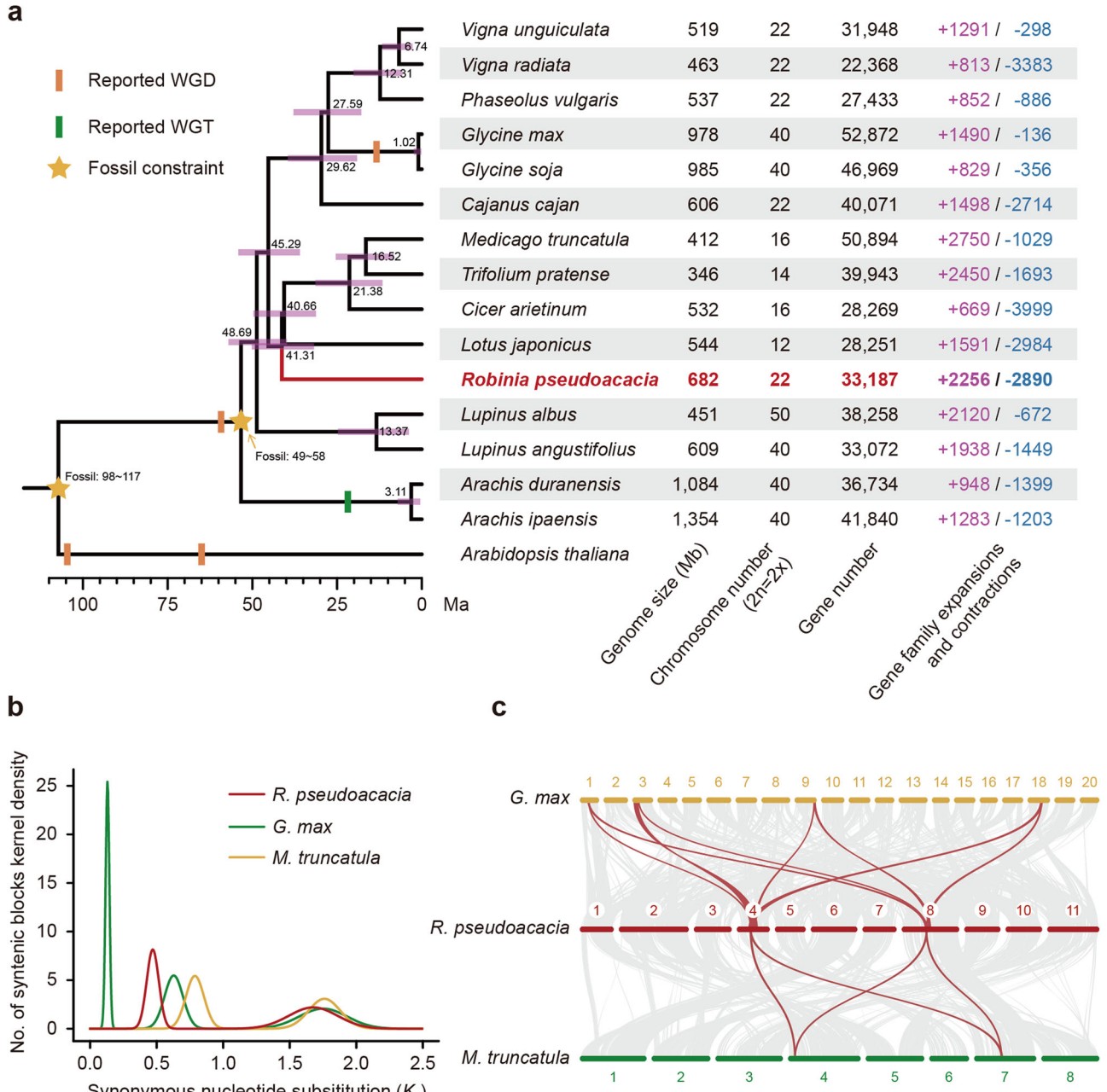

**Fig. 2 Comparative genomic analyses of the *R. pseudoacacia* genome. a** Phylogenetic tree for *R. pseudoacacia* and the other 14 species from Fabaceae, with *A. thaliana* as the outgroup. Bootstrap values for the nodes were 100. The estimated divergence time for each node is indicated with bars as the 95% confidence intervals (CI). The reported WGD, WGT, and the used fossil constraint are also labeled. Genome statistics for each species are shown to the right. **b** Ks distributions reveal WGD events during the evolution of *R. pseudoacacia*, *G. max*, and *M. truncatula*, respectively (Supplementary Data 3). **c** Collinear relationship between *R. pseudoacacia*, *G. max*, and *M. truncatula* is represented by a ratio of 4:2:2 (*G. max* : *R. pseudoacacia* : *M. truncatula*). Interspecific syntenic blocks that span more than 14,000 genes are indicated with lines, and some of the 4:2:2 blocks are highlighted in red.

*max*, and a 2:2 syntenic relationship between *R. pseudoacacia* and *M. truncatula* (Fig. 2c and Supplementary Fig. 4). A selected gene family was comprised of the same ratio of gene copies from different species analysed above, which also confirms these two WGD events (Supplementary Fig. 5). These results provide additional evidence that there was no *R. pseudoacacia*-specific WGD event.

**Gene family births and expansions implicated in wood formation.** During the long-term evolutionary history, the expansion of gene families is likely to lead to their functional diversifications

(e.g., neofunctionalization and sub-functionalization), and are further expected to contribute to dynamic adaptations that increase their survival in plants[35,36]. To understand the main reasons for the large gene set in *R. pseudoacacia*, we performed gene family analyses of the 16 plant species to reveal their contribution to adaptive divergence. A total of 31,508 *R. pseudoacacia* genes (94.9%) were clustered into 15,789 gene families where 612 gene families were specific to *R. pseudoacacia*. Our results showed 2256 gene families were expanded in the *R. pseudoacacia* genome and ranked third among all the analysed species (Fig. 2a).

From these comparisons, duplicated genes that have experienced functional diversifications are likely to reflect novel traits in

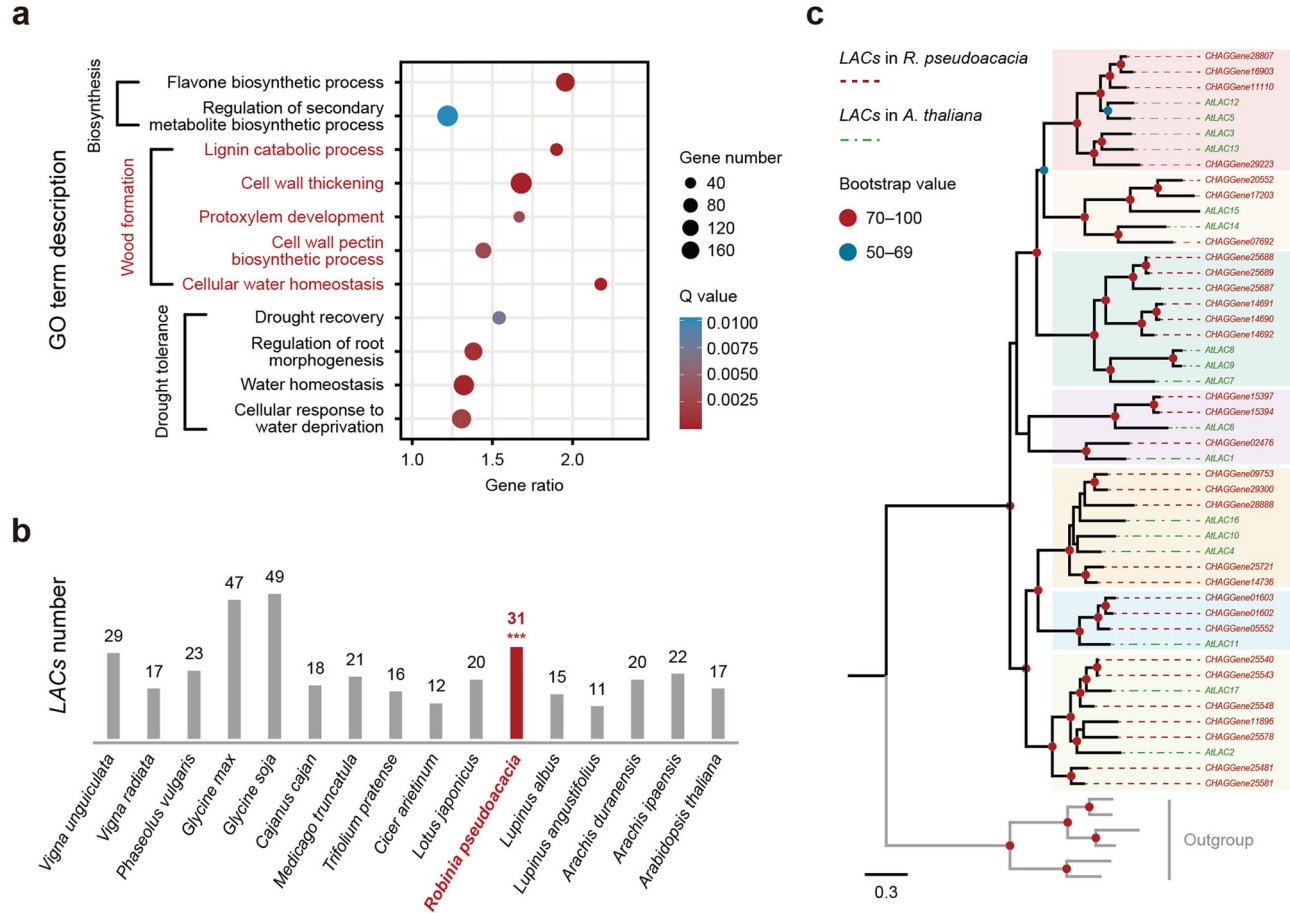

**Fig. 3 Gene family expansion of *R. pseudoacacia*. a** A number of 11 GO categories related to wood formation, biosynthesis, and drought tolerance are significantly enriched in the expanded gene families of *R. pseudoacacia* (all *P* < 0.05). **b** Gene numbers for the identified laccase (*LAC*) gene family between each species. The two-tailed one-sample Student's *t*-test was performed to examine the statistical significances, which are denoted with asterisks (*** *P* < 0.001). **c** Maximum likelihood (ML) tree based on the protein sequences of the *LAC* genes in *R. pseudoacacia* and *A. thaliana*. The analysis is performed under the PROTGAMMAGTR model with 100 bootstraps.

a species[37]. Our GO enrichment analysis found that the expanded gene families in *R. pseudoacacia* were significantly overrepresented in three major functional categories, including wood formation, biosynthesis, and drought tolerance (*Q* < 0.05, count >35; Fig. 3a and Supplementary Table 12), which likely contributed to the species-specific characteristics leading to their wide cultivation in urban greening. This result is in accordance with previous physiological observations in *R. pseudoacacia* and highlights the traits that rendered it an advantageous choice for urban planners. Specifically, *R. pseudoacacia* is visually attractive and economically competitive in urban planting because of its main mechanism of fast growth that then follows with nutrient accumulation and early death that then opens up a growth spot[38,39].

In this study, we found that the *laccase* (*LAC*) gene family was significantly (*P* < 0.05) expanded in the *R. pseudoacacia* genome. The laccase (*LAC*) gene family encodes enzymes that catalyze oxidation-reduction reactions and is wildly reported to be involved in lignin biosynthesis, which is essential for wood formation and improves the water transport capacity of plants[40]. Here, we found that there were 31 genes that belong to the *LAC* gene family in the *R. pseudoacacia* genome from manually inspecting these genes in 15 species, verifying their conserved Cu-oxidase domains (Cu-oxidase, Cu-oxidase_2, Cu-oxidase_3) and gene structures (Supplementary Data 1). We also found that *R. pseudoacacia* contained a significantly larger (*P* = 7.11 × 10$^{-7}$)

number (also the largest number) of *LAC* genes than all other analysed species in Leguminosae, except for *G. max* and *G. soja* which experienced an additional round of WGD (Fig. 3b). To reveal the evolutionary trajectories of these *LAC* genes, we further constructed the phylogenetic tree of the *LAC* gene family between *R. pseudoacacia* and *A. thaliana* (Fig. 3c and Supplementary Table 13). The phylogenetic tree clustered 48 *LACs* (31 from *R. pseudoacacia* and 17 from *A. thaliana*, respectively) into seven groups. Each group comprised 8, 5, 9, 5, 8, 4, and 9 members. We also observed a noticeable expansion of *LACs* for *R. pseudoacacia* in most groups, and the expansion of *LACs* could highlight their probable contribution to the stronger capacity of wood formation in *R. pseudoacacia*[41,42].

**Population genomic analyses of *R. pseudoacacia* populations cultivated in urban green spaces.** *R. pseudoacacia* is widely cultivated in Chinese cities, especially in central China. To explore the species-specific adaptative traits of *R. pseudoacacia* for urban planting, we sampled 59 *R. pseudoacacia* individuals from 14 counties across five provinces in China (Fig. 4a and Supplementary Table 14) with a median sampling size of five individuals for each county. The whole genomes of these 59 individuals were resequenced to an average depth of 14×. Approximately 98% of these re-sequencing data have been aligned to the reference genome and covers more than 90% of the genome assembly, which reflected the high quality of this re-sequencing dataset (Supplementary Table 15).

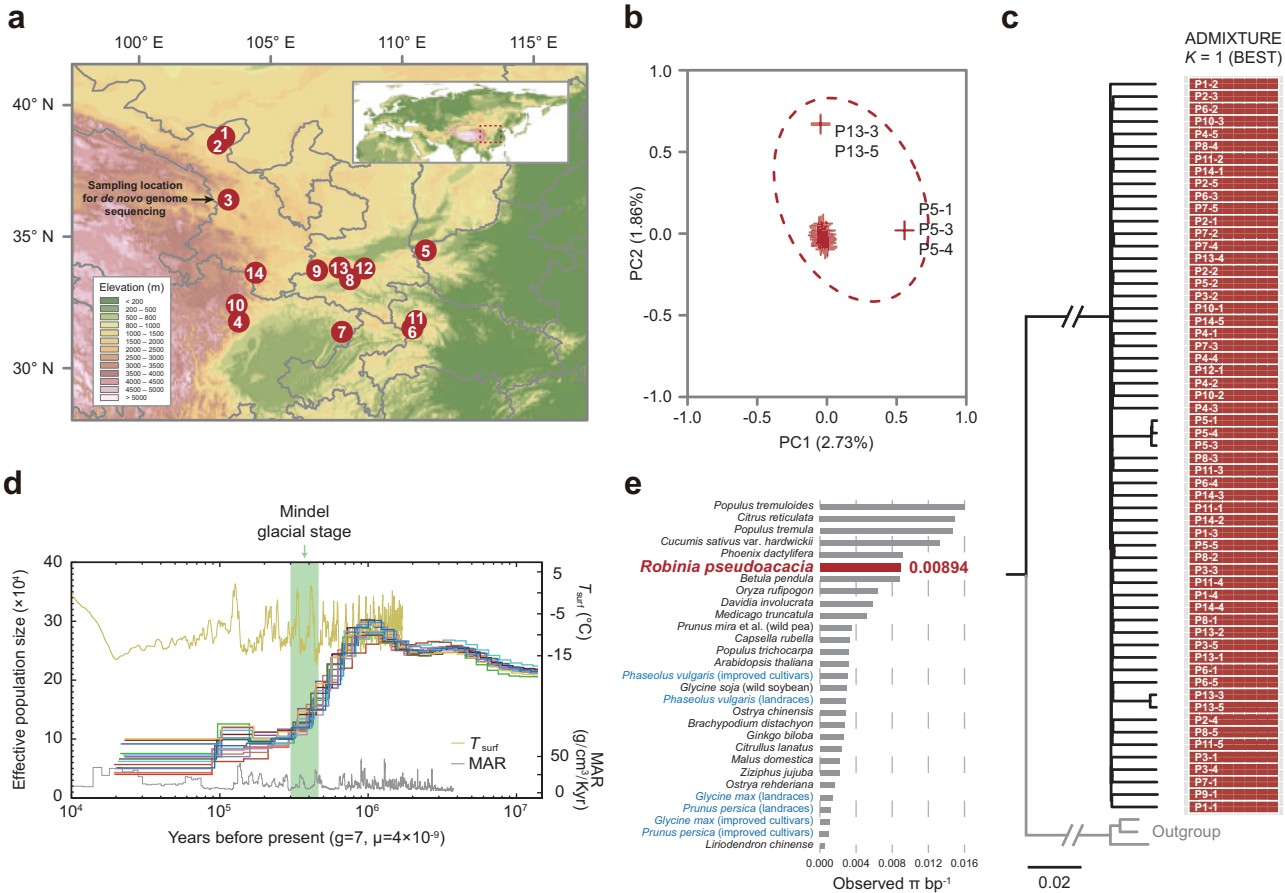

**Fig. 4 Population genomic analyses of *R. pseudoacacia*. a** Geographic distribution of the *R. pseudoacacia* samples sequenced in this study. The population IDs were labeled in the figure. **b** Principal component analysis (PCA). The numbers in brackets represent the fraction of variance explained by each component. The outlier individuals were labeled in black text. **c** Maximum likelihood (ML) tree and ADMIXTURE analysis. The phylogenetic analysis was performed based on the whole-genome SNPs of 59 *R. pseudoacacia* individuals, with individual IDs marked alongside the corresponding sections of the ADMIXTURE results. For each ID, the prefix before "−" corresponds to the sampling site number displayed in panel **a**, while the suffix after it indicates the individual number. Three *L. japonicus* individuals were used as the outgroup. **d** Demographic history inferred by the PSMC model. **e** Observed genome-wide sequence diversity (π) for *R. pseudoacacia* and other species. The species names of cultivars (including landraces) and wild lineages were labeled in blue and black colors, respectively.

We detected a total of 17,986,674 high-quality single nucleotide polymorphisms (SNPs) from these 59 individuals.

To identify the genetic structure of these samples, we performed a principal component analysis (PCA) based on their pairwise genetic distances that was calculated using autosomal SNP data. Critically, the PCs from 1–10 only explained a total of 14.15% of the genetic variance and could not separate individuals collected from the different locations (Fig. 4b and Supplementary Fig. 6). The two outlier groups were collected from two distinct counties, and were comprised of three individuals from the P5 population and two individuals from the P13 population, respectively (Supplementary Table 15). Surprisingly, they were not clustered in the same group with other individuals collected from the same or adjacent location.

To further examine this counterintuitive result, we evaluated the genetic relationships between these samples by establishing an ML phylogenetic tree. No elevated relatedness was found among the individuals sampled from the same city. In contrast, individuals collected from distinct cities were clustered together (Fig. 4c). Our phylogenetic tree also suggested that the clustering from the P5 group and the P13 group may have resulted from these samples coming from the same lineage, (i.e., they are children from the same parents and came from a similar plant nursery). Further justifications were also provided from an

Admixture analysis (Fig. 4c). Our results showed that $K = 1$ was the best modeling choice, and that the $K = 14$ modeling choice that divided *R. pseudoacacia* samples into local sub-populations based on sampling location was firmly rejected (Supplementary Figs. 7, 8). Our partial Mantel test, designed to evaluate the influence of isolation by distance (IBD) and isolation by environment (IBE) on genetic structure, revealed no significant correlation between geography or environment and genetic differentiation (Supplementary Fig. 9). These results confirmed that these *R. pseudoacacia* individuals were introduced through deliberate cultivation.

After these comparisons, we performed Pairwise Sequentially Markovian Coalescent (PSMC) analyses and found that the individuals collected from the different locations shared identical ancient demographic histories (Fig. 4d). Together, these results likely indicate that there has not been enough time to diverge from each other due to the fact that all *R. pseudoacacia* individuals in China were introduced by humans after the 1870s[43]. The estimated effective population size ($N_e$) of the *R. pseudoacacia* population reached its highest around 1 million years ago, followed by a sharp decline that coincides with the Mindel glacial stage.

Currently, most cultivated plants suffer from recent bottleneck effects and inbreeding introduced by human activities, such as

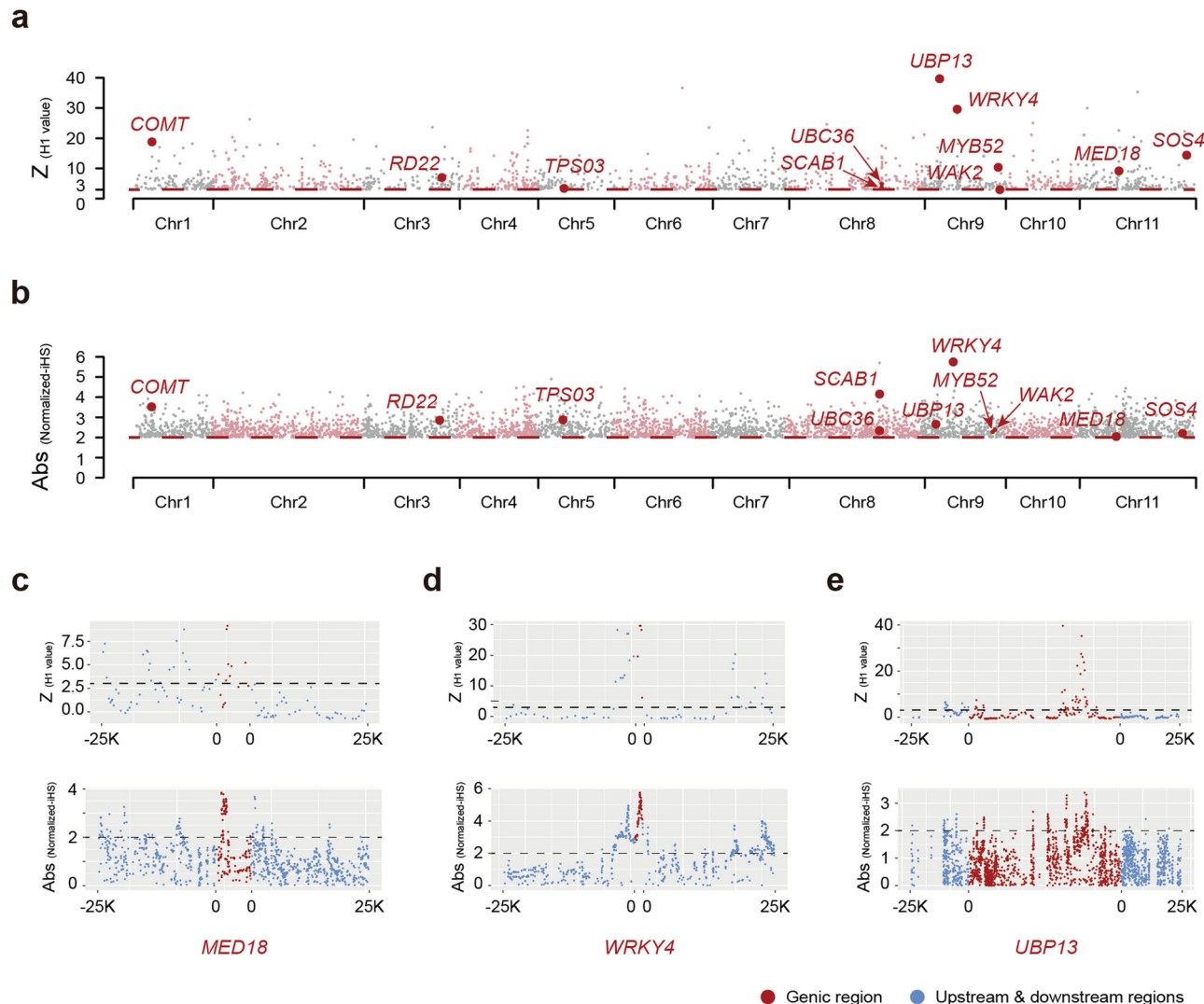

**Fig. 5 Genomic signatures of positively selected genes (PSGs). a, b** The candidate PSGs identified using H1 statistics and integrated haplotype scores (iHSs), respectively. Only the genes with significant values are shown. Eleven candidate PSGs are labeled with red circles and the gene name.
**c–e** Z-transformed H1 values and absolute values of the normalized iHS for *MED18* (**c**), *WRKY4* (**d**), and *UBP13* (**e**), respectively. **a–e** The threshold values (H1 value ≥3 and an absolute value of normalized iHS ≥2) were labeled in the figures.

domestication and intercontinental introductions, which might always reduce the genetic diversity[44,45]. Compared to populations with high genetic diversity, populations characterized by low genetic diversity are more susceptible to the detrimental effects of genetic drift and the accumulation of deleterious variations[46,47]. Therefore, maintaining sufficient genetic diversity is essential for species to adapt to changing environments. Here, we found that *R. pseudoacacia* displayed an observed diversity of $\pi = 8.94 \times 10^{-3}$, which was noticeably higher than other cultivated species and even higher than some wild plant populations (Fig. 4e and Supplementary Table 16). These highlight the abundance of *R. pseudoacacia*'s gene pool and their high potential to survive in changing environments.

**Genetic basis of the selected traits favored by urbanites.** To reveal the genetic basis of the traits favored by urbanites at the population level, we performed a comprehensive analysis of the whole genomes of 59 cultivated *R. pseudoacacia* trees planted in diverse environments and examined their genomic signatures for recent selective sweeps. Recent selective sweeps are usually reflected by long and unusually frequent haplotypes and high haplotype

homozygosity, which can be captured by iHS and H12 statistics, respectively[48]. Here, we analyzed the putative selective-sweep regions by detecting genetic regions that showed both iHS and H12 significance. These putative selective-sweep regions harbored a total of 615 positively selected genes (PSGs), indicating their potential role involved in the adaptation of *R. pseudoacacia* to urban environments (Fig. 5a, b and Supplementary Data 2). To further understand the evolutionary forces that accompany urban planting, Gene Ontology (GO) enrichment analyses were used to summarize the biological function of the gene set located in these candidate regions under recent selection. These analyses revealed several functional categories with enriched signals for selection that included cell wall thickening (GO:0052386), sporopollenin biosynthetic processing (GO:0080110), root meristem growth regulation (GO:0010082), and bacterium detection (GO:0016045) (Supplementary Table 17). The candidate genes showed several key characteristics favored in urban planting.

In urban plant community assemblies, the characteristics of tree flowers are one of the most important traits because they improve the aesthetics of urban streets and green spaces[49,50]. In this study, two genes for flower development were categorized to

be under selection and were consistent with the unique traits observed in *R. pseudoacacia*, which suggests that these genes may be involved in the breeding processes that led to their extensive urban planting in China. For example, *TPS03* encodes a catalysis enzyme that plays an important role in monoterpene synthase that may contribute to the appealing fragrance of tree flowers acclaimed by urbanites. Additionally, *MED18* (Fig. 5c) encodes for the subunit of the head submodule for the plant mediator complex and has been previously shown to regulate flowering time and alter floral organ number[51].

Compared to non-urban areas, municipal green spaces usually have a more shallow fertile soil layer and a reduced water retention capacity[52–54], which bring distinct challenges to trees planted in urban areas. Here, we found that the *SOS4* gene that is essential for root hair development, showed notable signatures for recent selection, as well as some genes related to water utilization, such as *WRKY4*, *SCAB1*, and *RD22* (Fig. 5d). Specifically, *WRKY4* and *MYB52* play important roles in drought stress adaptation[55,56]. *SCAB1* is reported to encode an actin-binding protein that controls stomatal movement, which is consistent with previous observations that *R. pseudoacacia* can adapt to drought conditions by reducing transpiration[57]. Additionally, *RD22* is a drought-responsive gene that have been found to be differentially expressed during water deficit[58,59]. The evolution of these genes at the population level may be beneficial to increase the fitness of trees planted in green spaces, especially given that *R. pseudoacacia* is usually shallow-rooted.

Along with these genes, three other genes related to the immune system were also found to show footprints of recent selection. For example, *UBP13* (Fig. 5e) encodes a ubiquitin-specific protease that is responsible for initial pathogen perception[60], and *UBC36* encodes an E2 ubiquitin-conjugating enzyme involved in dampening immune signaling[61]. The wall-associated receptor-like kinase gene, *WAK2*, also plays an important role in disease resistance[62]. Together, these genes influence the immune system and may reflect the fact that plants in cities face increased susceptibility to disease caused by different pathogens and pollutants compared to the ones living in the wild[63]. Overall, our findings highlight the genetic characteristics favored in urban planting and shed light on the evolutionary forces that accompany the process of urban greening.

**Conclusions**. In this study, we took advantage of cutting-edge genomic methods to reveal the genomic basis and consequences of the wide distribution of *R. pseudoacacia* in urban green spaces in China, and provide novel insights into urban greening. We first generated an annotated chromosome-level genome assembly by combining Illumina short-read sequencing, nanopore long-read sequencing, and chromosomal conformational capture (Hi-C) technologies to create the first reference genome for *R. pseudoacacia* (Fig. 1). Our following comparative genomic analyses showed that gene families related to traits favored by urbanites were noticeably expanded, and include wood formation, biosynthesis, and drought tolerance (Figs. 2, 3). We additionally surveyed 14 cities, collected 59 *R. pseudoacacia* individuals, and sequenced the whole genomes of from these samples to further reveal how genetic and phenotypic effects accompany the urban planting process introduced by urbanization (Fig. 4a).

Our comprehensive genetic structure and demographic analyses strongly indicate that the presence of *R. pseudoacacia* in urban green spaces is a result of deliberate cultivation rather than natural dispersal (Fig. 4b, c and Supplementary Fig. 9). Additionally, *R. pseudoacacia* planted in cities showed a rich genetic diversity of $\pi = 8.94 \times 10^{-3}$ (Fig. 4e) that is distinct from most cultivated plants, which suffering from recent bottleneck effects and inbreeding. Together, these effects highlight their potential to withstand environmental disruptions, which is favored by city

planners. In addition to stress-resistance, esthetics is another major consideration for urban residents. Such characters have also been recovered in our selective-sweep analyses. We find genes play important roles in the biological process related to flower development, water retention, and immunization that showed very recent selection in their genomic signatures, which suggests that the massive urban planting process accompanies substantial genetic effects in plant genomes (Fig. 5).

Due to our extensive long-read sequencing, we were able to uncover the full genomic content of *R. pseudoacacia* that allows for the correct identification of many unknown structural variations and repetitive elements that will be a valuable resource for future genetic diversity studies in plant taxa. Our resources and results reported provide novel insights into the construction of urban green infrastructures at comparative and population genomic levels, as well as valuable foundations for the agronomic understanding and molecular breeding of *R. pseudoacacia* in the future.

Still, our current study only focused on SNPs with a relatively modest sample size. Although the current sampling is enough to explore the species-specific adaptative traits of *R. pseudoacacia*, further studies that involve a wider range of sampling that represent different environmental stressors combined with transcriptomic and methylation data, as well as structural variation data, are needed to enhance our knowledge on urban greening that facilitate its improvement. Additionally, *R. pseudoacacia* could use asexual proliferation. During the clonal evolution process, somatic mutations are a major source of genetic diversification[64]. Somatic mutations in citrus led to a broad spectrum of phenotypes, including changes in fruit shape, color, acidity, maturation season, developmental changes related to sterility, flowering time, and tree architecture[65]. Therefore, the somatic mutation rate and pattern should be considered in future studies. Ultimately, the comprehensive evaluation of data obtained from genomic, physiological, and ecological studies will become increasingly important in applied contexts that target the planting of trees in urban green spaces.

## Methods

**Plant materials and genome sequencing**. We collected one *R. pseudoacacia* individual (Fig. 1a) at Lanzhou University, Gansu Province, China (36°2′57″N, 103°51′34″E). One Oxford Nanopore Technologies (ONT) library, one Illumina paired-end sequencing library, one Hi-C library, and three RNA-sequencing libraries were prepared based on this individual. Specifically, we extracted high molecular weight DNA from the leaves of this individual, established an Oxford Nanopore Technologies (ONT) library, and sequenced it on the ONT PromethION platform according to the manufacturer's instructions. We then supplemented these long reads with additional high-accuracy short reads from one Illumina paired-end sequencing library. We extracted DNA from leaves of the same individual using the CTAB method[66,67], after which paired-end sequencing libraries were established, PCR amplified, and sequenced on an Illumina NovaSeq 6000 platform following the workflow recommended by Illumina. To obtain chromosome contact information and achieve chromosome-scale contiguity, we extracted DNA from the same individual, and constructed Hi-C libraries following the standard protocol described previously[68], in which a 4-cutter restriction enzyme, DpnII, was used for digestion. We amplified these Hi-C sequencing libraries and sequenced them on the Illumina NovaSeq 6000 platform. To aid the gene prediction and annotation, flowers, stems, and leaves from the same individual were used for RNA sequencing. In brief, we extracted total RNA from these three tissues using TRIzol and RNA purification kits, and prepared three Illumina mRNA TruSeq libraries, respectively, following the manufacturer's guides. RNA sequencing was performed on the Illumina NovaSeq 6000 platform.

**Chromosome-level genome assembly**. After performing base-calling and read filtering, we corrected the per-base accuracy of the PromethION long reads (~110×) and assembled them into contigs using NextDenovo v2.3 (https://github.com/Nextomics/NextDenovo) (-task all -parallel_jobs 20 -read_cutoff 1k -genome_size 680 M). To further polish this contig-level assembly, ~111 Gb (~162×) of high-accuracy Illumina paired-end reads were generated. To remove allelic haplotigs, we utilized Purge Haplotigs v1.1.1[69], resulting in the final contig-level assembly. Based on these high-accuracy short reads, we fixed base errors in the

contig-level assembly using NextPolish v1.2.2[70]. These data were also used to perform k-mer analyses and estimate the genome size of *R. pseudoacacia*[71]. We then evaluated, filtered, and mapped the paired-end Hi-C reads (~114×) to this contig-level assembly using HiCUP v0.8.0[72]. The resultant BAM file was then processed with ALLHiC v0.8.12[73], which built a chromosomal-scale assembly with the contig-level assembly and the Hi-C data using the innovative "prune", "partition", "rescue", and "optimize" steps.

**Repeat annotation**. We performed repeat and gene predictions following the customized pipeline previously described in Pascoal et al. with several modifications[74]. Briefly, we first constructed a de novo repeat library for *R. pseudoacacia* using RepeatModeler v2.0[75] with RECON v1.08[76] and RepeatScout v1.0.6[77]. For repetitive element identification, we used BLAST+ v2.2.31[78] and RepeatMasker v4.1.0[79] to search the *R. pseudoacacia* genome against our de novo repeat library and the Repbase database v23[80]. LTR_retriever v2.8.7[81], which integrates LTRharvest[82], and LTR Finder v1.07[83] was then implemented to predict long terminal repeat retrotransposons (LTR elements). After, the repeat identification results from the different software packages were integrated and redundancy was eliminated to produce the final repeat annotation. We calculated the genetic distance (K) between the 5′ LTR and 3′ LTR sequences using DnaDiSt, a program within PhyliP v3.696 (Felsenstein, 2004). To estimate the insert time (T) of each LTR, we used the formula $T = K/2r$, where r represents the nucleotide substitution rate estimated by BASEML in the PAML package v4.9j[84].

**Gene prediction and functional assignment**. After repeat-masking the *R. pseudoacacia* genome, transcriptome-based, homology-based and ab initio predictions were performed. For transcriptome-based prediction, we used Trinity v2.6.6[85] (-seqType fq -max_memory 50 -CPU 20) to obtain a de novo *R. pseudoacacia* transcriptome assembly based on our RNA-Seq data. We then processed this transcriptome assembly using Program to Assemble Spliced Alignments (PASA v2.3.3)[86]. By mapping it to the reference genome assembly, PASA predicted open reading frames (ORFs) and gene structures. We also aligned these trimmed RNA-Seq reads to the *R. pseudoacacia* genome assembly using HISAT2 v2.1.0[87]. After, an intron hint file was generated using the bam2hints function in AUGUSTUS v3.3.3[88]. We used these results to train AUGUSTUS to perform ab initio gene prediction. For homology-based prediction, we aligned protein sequences of seven plant species from Phytozome v13[89] (https://phytozome-next.jgi.doe.gov) (*A. thaliana, C. arietinum, Glycine soja, Lupinus albus, Populus trichocarpa, T. pratense,* and *Vigna unguiculata*) to the repeat-mask the *R. pseudoacacia* genome using TBLASTN (E < 10−5). The resultant candidate protein-coding regions were then refined and further processed by GeneWise v2.4.1[90] to generate a homology-based gene set with accurate splice junctions. All gene sets predicted by ab initio, homology, and transcriptome-based methods were passed into EVidenceModeler v1.1.1[84] to produce a consensus gene set. We further upgraded this consensus gene set by predicting untranslated regions (UTRs) and alternatively spliced isoforms based on our transcriptome data, which was implemented in PASA v2.3.3. We used BUSCO v4[91] with embryophyta_odb10 database[92] to evaluate the completeness of the genome assemblies (-l embryophyta_odb10 -m genome -c 10 -e 1e-03) and the official gene set (-l embryophyta_odb10 -m proteins -c 10 -e 1e-03), respectively. Circos v0.69[93] then was used to visualize these genomic metrics in a circular layout.

Functional annotations from this consensus official gene set were obtained based on Swiss-Prot (version 2020_04)[94], TrEMBL (version 2020_04)[94], NCBI non-redundant protein (NR, release 20200502)[95], and InterPro v84 databases[96]. Specifically, we ran BLAST+ with a cut-off E value of 1E-05 and a maximum target sequence number of 20 to obtain the best-hits (-evalue 1e-5 -num_threads 30 -max_target_seqs 20), and assign descriptors of these best-hits to the predicted transcripts, respectively. InterProScan v.5.28[97] was used to retrieve functional domains and Gene Ontology (GO) annotations based on the InterPro database. Additionally, we used Blast2GO[98] to further assign GO terms to the genes that were not annotated by InterProScan. For the Kyoto Encyclopedia of Genes and Genomes (KEGG) annotation[99], we aligned the protein sequences of the *R. pseudoacacia* gene set against the KEGG (family_eukaryotes) database, assigned KEGG Orthology (KO) terms, and reconstructed pathways by submitting these sequences to the KEGG Automatic Annotation Server (KAAS)[100]. We then implemented clusterProfiler package v4.1.4[101] to perform GO enrichment tests.

**Comparative genomic analysis**. We identified orthologous genes among 16 plant species by performing an ortholog clustering analysis implemented in OrthoFinder v2.3.8[102] (-S blast -M msa -t 50 -T fasttree -A mafft). The general phylogeny was then resolved by performing RAxML v8.2.12[103] analysis, which constructed a whole-genome maximum likelihood (ML) phylogenetic tree for these species based on their concatenated four-fold degenerate sites (4DTv) under the GTRGAMMA model. The outgroup was *A. thaliana,* and the analysis was performed with 100 bootstraps for robustness. We then estimated the divergence time of each node in this phylogenetic tree by running MCMCtree program in the PAML package v4.9j[84] with two fossil constraints (*Arachis duranensis* and *M. truncatula* [49–58 million years ago (Ma)], as well as *Glycine max* & *A. thaliana* [98–117 Ma]) acquired from TimeTree (http://www.timetree.org/). Following these general phylogeny analyses, we passed the results

obtained from the OrthoFinder pipeline to CAFE v4.2[104] to test the patterns in gene family evolution (expansion/contraction).

To further explore the molecular mechanism of wood formation in *R. pseudoacacia,* we downloaded the sequences of the laccase (*LAC*) gene family for *A. thaliana* (Supplementary Table 11) from the TAIR database (https://www.arabidopsis.org/) and identified the corresponding *LACs* for the other 15 species. We first used BLASTP v2.2.29+ to search for the candidate orthologous to *LACs* against the *A. thaliana LACs.* HMMER v3.2.1 (http://hmmer.org/)[105] was then employed to scan the domain information for each candidate gene. Only the genes that have three conserved Cu-oxidase domains (Cu-oxidase, Cu-oxidase_2, and Cu-oxidase_3) were retained. We also detected and retained the candidate genes with the four copper binding regions (HxH, HxH, HxxHxH, and HCHxxxH). A phylogenetic analysis was then carried out by RAxML to examine if the *LAC* candidate genes were clustered together.

**Identification of whole-genome duplication (WGD) events and genome synteny**. Inter- and intragenomic homologous genes were identified using ColinearScan[106]. We first used BLASTP v2.2.29+ (E value <1 × 10−5) to search for the putative paralogous and orthologous gene pairs within or between genomes with a maximum of 20 alignments for each query sequence. The maximal collinearity gap length between genes was set to 50 for the ColinearScan. The synonymous substitution (*Ks*) values of the identified colinear gene pairs were then calculated using the YN00 program in the PAML package using the Nei–Gojobori method[107]. The median *Ks* value for the colinear gene pairs from each colinear block was shown in the syntenic dot plots to help distinguish event-related syntenic regions. Gaussian kernel density fitting was then performed to estimate the probability density distribution of inter- and intraspecies *Ks* values with the bins number set to 200. The above synteny analyses were performed using the wgdi toolkit (https://github.com/SunPengChuan/wgdi)[108].

**Sampling and whole-genome re-sequencing of *R. pseudoacacia* populations**. To explore the species-specific evolutionary traits of *R. pseudoacacia,* we surveyed 14 counties across five provinces which covered arid, humid, plateau, and plain areas, to collect population genomic data. In total, 59 individuals were sampled. DNA was extracted from the leaves of these individuals using the CTAB method. We used a Nanodrop, 1% agarose gel electrophoresis, and Qubit to check the quality and purity of the extracted DNA. Illumina paired-end libraries were then prepared for each individual, following the manufacturers' laboratory protocols, and sequenced on the Illumina NovaSeq 6000 platform. We also downloaded three *L. japonicus* samples as the outgroup (NCBI accession number: SRR11495342, SRR11495343, and SRR447704). For raw reads, Scythe (https://github.com/vsbuffalo/scythe) and Sickle (https://github.com/najoshi/sickle) were used to remove the adapter sequences and filter out the low-quality sequences (quality score <20), respectively. We then mapped these clean reads to the chromosome-level *R. pseudoacacia* genome using BWA v0.7.17[109].

**SNP genotyping for phylogenetic and population structure analyses**. We genotyped the 62 individuals (59 individuals of *R. pseudoacacia* and three individuals of *L. japonicus*) following the GATK best practices[110]. Before the samples were genotypes, Samtools v1.10[111] and Picard v2.23.6 (http://broadinstitute.github.io/picard/) were used to sort and format the resulting files from BWA, and to remove PCR duplicates. For the GATK v3.8 best practice, the HaplotypeCaller function was used to call SNPs and InDels via local re-assembly of haplotypes. The genetic variants detected from the same population were then merged by the CombineGVCFs function to expedite downstream processes. We used GenotypeGVCFs to perform the accurate re-genotyping based on these merged gVCF files. A set of hard filtering criteria was decided based on density plots of the raw SNP dataset (QD <2.0| |MQ <40.0 || ReadPosRankSum <−8.0||FS >60.0 || HaplotypeScore >13.0 || MQRankSum <−12.5||SOR >3). In order to generate a robust set of SNPs, we additionally removed low-quality SNPs with abnormally low depth (<2 × or 1/5 average depth of the corresponding sample), extremely high depth (>50× or fivefold average depth of the corresponding sample), and low-quality scores (<30). Furthermore, we discarded SNPs that were labeled as indels and located within 5 bp of an InDel. If the genotype information was missing from most of the individuals (>13 *R. pseudoacacia,* >1 *L. japonicus*), the site was treated as unknown.

The generated VCF file was then used by Vcftools v0.1.16[112] to calculate genome-wide genetic diversities. An ML phylogenetic tree was established based on whole-genome SNPs by RAxML v8.2.12[103], and a reduced SNP set without SNPs in high linkage disequilibrium (LD) between each sample, was generated by PLINK v1.07[113], which was based on their pairwise LD information. We then assessed the population structure of the sampled individuals by passing this reduced SNP set to ADMIXTURE v1.3.0[114] with plausible numbers for the ancestral populations (K) from 1 to 14. The best module was chosen based on cross-validation error rates. To further confirm the population structure, the smartpca program in EIGENSOFT v7.2.1[115] and ggplot2 package in R were used to perform principal component analysis (PCA) to visualize their results, respectively. We then performed a series of Pairwise Sequentially Markovian Coalescent analyses using PSMC v0.6.5-r67[116] to reconstruct the historical trajectories of changes in effective population sizes of *R. pseudoacacia.* To convert the scaling time and population size to actual values, we

applied a generation time of 7 years and a mutation rate of $4 \times 10^{-9}$ per nucleotide per year. These conversion factors allowed us to estimate the time and size in real-world terms.

**Effects of IBD and IBE on genetic structure**. We evaluated the effects of isolation by distance (IBD) and isolation by environment (IBE) on genetic variation by Mantel tests. First, we calculated pairwise geographical distances between sampling sites using the geographical coordinates with the R package Geosphere (https://github.com/rspatial/geosphere). Secondly, we calculated the environmental distance (Euclidean distance) among the 16 populations based on the environmental variables described in the "Environmental variables" section using the dist function in R software. To measure genetic distance, we utilized Arlequin v3.5[117]. Finally, we conducted partial Mantel tests with 9,999 permutations using the Vegan v.2.5-7 package[118] in R to test the relationship between geographical/environmental distances and genetic distances.

**Selective-sweep analyses**. We used a combinatorial approach to ensure robust inferences for selection. Briefly, we used Beagle v5.4[119,120] to reconstruct haplotypes from unphased SNP genotype data and impute missing data. The SelectionHapStats[121] program (with the parameters: -w 50 -j 20) was then used to examine frequencies from multiple haplotypes between each other and to identify both hard and soft selective sweeps and distinguish between these two types of selection sweeps based on H12 and H2/H1 statistics. Additionally, we applied Selscan v2.0.0[122] to calculate the integrated haplotype score (iHS), which is designed to detect unusual haplotypes around a particular SNP compared to the whole genotype. Subsequently, the metrics detected by these methods were synthetically evaluated to reveal robust selective sweeps. The genes with a significant H1 value ($Z \geq 3$) and a significant iHS value (absolute value of normalized iHS ≥2) were then identified as the positively selected genes.

**Statistics and reproducibility**. Statistical analyses (the two-tailed one-sample Student's *t*-test and partial Mantel test) were performed using R (v4.1). The *P* values associated with gene family sizes were computed by CAFE[104]. The *Q* values for functional enrichment tests were calculated using clusterProfiler[101]. In population genomic analyses, we sequenced three to five individuals for each population, except for two populations, each of which only had one sample in the wild.

**Reporting summary**. Further information on research design is available in the Nature Portfolio Reporting Summary linked to this article.

## Data availability

Numerical sources can be found in Supplementary Data 3 for Fig. 2b and Supplementary Table 16 for Fig. 4e, respectively. All sequencing reads that support the findings of this study have been deposited in the Genome Sequence Archive (https://ngdc.cncb.ac.cn/gsa/) under project number PRJCA011483. The genome assembly file, all the annotation files, and source data for phylogenetic and population analyses are available at Figshare (doi.org/10.6084/m9.figshare.23301668). All other data are available from the corresponding authors on reasonable request.

## Code availability

The custom scripts and analysis pipelines have been made accessible to the public via GitHub repositories (https://github.com/myBioFun/ORTHO2TREE).

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

## Acknowledgements

We are grateful for the editors' and three reviewers' excellent suggestions to improve the manuscript. This study was supported by the National Natural Science Foundation of China (grant no. 32001085) and Fundamental Research Funds for Central Universities (grant no. lzujbky-2020-34).

## Author contributions

D.R., X.Z., Z.W., S.W., and B.L. conceived the project and designed the analyses; Z.W. conducted the research; Z.W., X.Z., H.Z., W.L., S.W., B.L., and D.R. analyzed the data; X.Z., Z.W., and D.R. wrote the paper; and all authors revised and approved the final manuscript.

## Competing interests

The authors declare no competing interests.
