## [Peer Review File · Communications Biology]

Reviewers' comments:

Reviewer #1 (Remarks to the Author):

In the manuscript entitled "Chromosome-level genome assembly and population genomics of *Robinia pseudoacacia* (black locust) provide insights into urban greening", the authors generated a high-quality, chromosome-level assembly for black locust (*Robinia pseudoacacia* L.), one of the most widely planted woody species in the world, and annotated its protein-coding genes and repetitive sequences. The reason for the relatively large genome size of black locust, compared to closely related species *C. arietinum* and *L. japonicus*, was explained. Comparative genomic studies revealed that gene families related to wood formation, flower development and drought tolerance in black locust adapted to urban greening requirements are expanded in the genome. This finding is interesting, but it is important to clarify whether the expansion of these gene families in black locust is the result of adaptation to urban environments or whether the black locust themselves have evolved such characteristics to become more popular in urban environments, since the authors also mention that all black locust trees in China were introduced by humans after the 1870s. The analysis of the population clarified that the distribution of black locust in different provinces and cities is more inclined to introgression than natural distribution, which is a positive result. The study also identified recently selected regions in the genome of black locust, which contain genes related to resistance to urban environmental disturbances, flower fragrance, drought resistance, and wood development, which should be useful for urban greening species improvement and breeding. Overall, the high-quality genome assembly of black locust still provides a valuable genetic resource for urban species adaptation and greening improvement and the data will be incredibly useful to the community.

Major Concerns:

1. The logic of the presentation of the results is problematic in the WGD section. The authors first briefly mention WGD from line 151, and then further examine WGD in the next section, where the two parts should be combined.

2. The authors predicted a much high number of genes in *R. pseudoacacia* than in *C. arietinum* and *L. japonicus*, which they claimed to be a result of gene expansion. However I am curious whether this is the result of redundancy of haplotype sequences due to the assembly process?

3. The authors mentioned that *R. pseudoacacia* exhibited a comparable ratio of genomic repeats to closely related species, which supports the idea that the large size of the black locust genome may be driven by the explosion of repetitive elements. However, it should be pointedly pointed out in the article that although the genomic repeats ratio was comparable, the total length of repetitive sequences may be greater in *R. pseudoacacia*. Additionally, how the exact number was between *R. pseudoacacia*, *C. arietinum*, and *L. japonicus*, and whether the timing of the outbreak of *R. pseudoacacia* LTRs (since the authors also mention LTRs as the main type of repetitive sequences) could be added to the analysis to support this conclusion. This would provide the reader with a more convincing and clearer understanding of genomic repeats of *R. pseudoacacia*.

4. A statement of the threshold values should be added to the legend of Figure 5A/B, otherwise the reader can't clearly understand the figure notes of only the genes with significant values are shown.

5. "Data acquired in this Whole Genome Shotgun project has been deposited in the BIG Data Center (<http://bigd.big.ac.cn>) under project number PRJCA011483."

I checked the BIG database but did not find any information related to the PRJCA011483 project in my search. If your manuscript is accepted, please make sure to publish all the raw data of sequencing.

Minor Concerns:

1. Fig. 1B: Please use the same style in this table in a standardized way, either with or without parentheses.

2. Legend of Fig 2C: Legend of Fig 2C: what is 4:2:2? Please add the detail. From the figure I can

see it's *G. max* : *R. pseudoacacia* : *M. truncatula*. Please indicate in the legend the specific object of this scale.

3. Line 102-103: The authors mentioned the resulting assembly showed better continuity than other currently available genomes from the family Fabaceae, but I am not aware of any evidence to support this result. At least I can't find relevant evidence in Fig. 2A. It is suggested that N50 information of the other genomes be included in Fig. 2A to support this conclusion.

4. Line 173-175: The term "closely related species" used here is ambiguous. How exactly is "closely related" defined in this context? Is it the extended gene set of *R. pseudoacacia* larger than that of all closely related species? Without evidence to support this, a clear description of *C. arietinum* and *L. japonicus* is suggested.

5. Line 213-216: The reader cannot get exactly which individuals are outliers from the supplementary table, and it is recommended to mark them in the Table S13.

6. Fig. 4C: The clustering of the different cities' individuals cannot be obtained intuitively from the Fig. 4C, and the individuals that make up the trees are not visible from the trees, which makes it difficult to understand the clustering and grouping. At least for me, I am unable to find this information from Fig. 4C or the supplementary table.

7. Fig. 4E, Table S14: I could not get information from the Fig. 4E whether other species were cultivars or not. The description of the results here is overly simplistic and crude, and I did not even get valid information on cultivars and non-cultivars. Although the genetic diversity of *R. pseudoacacia* are relatively high in the results, the lack of information on cultivars and non-cultivars questions about the validity of the findings here.

Reviewer #2 (Remarks to the Author):

Wang et al., reported a whole genome sequence for *Robinia pseudoacacia* and re-sequenced the species across China. The paper is well organized and the quality of the genome is quite good. The weakness part is the population genomic analysis. The authors should describing the re-sequenced pops/individuals in detail. When and where all these individuals from? How its genotype distributed and genetic structure of the introduced pops? Do the genotype associated with the environment factors?

Comments in detail:

L2: title: what insights? It is better to report the main conclusion directly in the title.

L24: what comparative analysis?

L42: "major considerations" is unclear here

L53: describe the introduced history in detail and cite relevant references.

L58-59: it is better not mix common and latin name.

L62: Chinese cites, very vague description.

L133-139: To me, this paragraph is empty, without useful information, can be used in any plant species.

L238-241: genetic diversity is not direct linked to fitness.

L247: reorganize the sentence.

Reviewer #3 (Remarks to the Author):

Robinia pseudoacacia is an important street trees in the world, with high ecological and economic values. In this study, Wang et al. tried to explore its genetic characteristics for urban greening. Firstly, the authors reported a high-quality chromosome-level genome assembly of *R. pseudoacacia*, and demonstrated the genetic mechanism for its characteristic traits (especially

wood formation) via comparative genomic analyses. Then, the population genomic analyses were carried out using 59 resequenced individuals collected from 14 counties. A total of 11 positively selected genes were further identified likely contributing to the urban greening-related traits. In summary, this research provides insights into the molecular mechanisms for urban greening, which is relatively novel in my opinion. The genetic resource provided in this research is also valuable for the future related studies. I think this MS is suitable for Communications Biology. However, there are still several issues that need to be addressed.

Major concerns:

1. The flower is a clear indication to recognize the *R. pseudoacacia*, which should be provided in Figure 1.
2. The authors claimed LAC gene family was expanded in *R. pseudoacacia*, which may be involved in the process of wood formation. I wonder that whether this gene family suffers natural selection? Whether they are identified as the positively selected genes? If not, why?
3. Gene families related to wood formation have been enriched in Figure 3A. These expanded families related to wood formation is important for a deeper understanding of black locust and should be provided in supplementary file.
4. More details should be provided in the Methods section, such as the program parameters, so that we can better assess the accuracy of results and follow the methods of this research.
5. L58-59: The Latin name of white clover should be provided.
6. L212: Please reconfirm whether PCs 1-10 could only explain a total of 14.15% genetic variance.
7. L353 and somewhere else: in "110x", the letter "x" should be changed to a multiplication sign.
8. Figure 2C: The color of LACs in *R. pseudoacacia* and *A. thaliana* is too similar to distinguish.
9. Fig. 4A: The sampling location of individual sequenced for de novo genome should be labeled in the figure.
10. Fig. 4D: Please describe how to set the generation time (g) and mutation rate (μ) in the Methods section.

Reviewers' comments:

Reviewer #1 (Remarks to the Author):

In the manuscript entitled “Chromosome-level genome assembly and population genomics of *Robinia pseudoacacia* (black locust) provide insights into urban greening”, the authors generated a high-quality, chromosome-level assembly for black locust (*Robinia pseudoacacia* L.), one of the most widely planted woody species in the world, and annotated its protein-coding genes and repetitive sequences. The reason for the relatively large genome size of black locust, compared to closely related species *C. arietinum* and *L. japonicus*, was explained.

Comparative genomic studies revealed that gene families related to wood formation, flower development and drought tolerance in black locust adapted to urban greening requirements are expanded in the genome. This finding is interesting, but it is important to clarify whether the expansion of these gene families in black locust is the result of adaptation to urban environments or whether the black locust themselves have evolved such characteristics to become more popular in urban environments, since the authors also mention that all black locust trees in China were introduced by humans after the 1870s.

Response: We thank the reviewer for this good question. The expanded gene families in *Robinia pseudoacacia* were analyzed and identified by comparing with multiple species. As we mentioned in the manuscript, all black locust trees in China were introduced and further planted for urban greening in less than 200 years. It is really a slim chance for the gene families to expand in such short time-span. Thus, the expanded gene families were the characteristic-specific to *R. pseudoacacia*, leading to their wide cultivation in urban greening. We have added the related statement in the revised manuscript (lines 179-185).

“Our GO enrichment analysis found that the expanded gene families in *R. pseudoacacia* were significantly overrepresented in three major functional categories, including wood formation, biosynthesis, and drought tolerance ($Q < 0.05$, count >35 ; Fig. 3A, Table S12), which likely contributed to the species-specific characteristics of *R. pseudoacacia*, leading to their wide cultivation in urban greening. This result is in accordance with previous physiological observations in *R. pseudoacacia* and highlights the traits that rendered it an advantageous choice for urban planners.”

The analysis of the population clarified that the distribution of black locust in different provinces and cities is more inclined to introgression than natural distribution, which is a positive result. The study also identified recently selected regions in the genome of black locust, which contain genes related to resistance to urban environmental disturbances, flower fragrance, drought resistance, and wood development, which should be useful for urban greening species improvement and breeding. Overall, the high-quality genome assembly of black locust still provides a valuable genetic resource for urban species adaptation and greening improvement and the data will be incredibly useful to the community.

Response: Thank you for your positive feedback on our study. We are pleased to your comment on our conclusions that black locusts were introduced to the sampled locations through cultivation rather than natural dispersal and find our research to be valuable.

Major Concerns:

1. The logic of the presentation of the results is problematic in the WGD section. The authors first briefly mention WGD from line 151, and then further examine WGD in the next section, where the two parts should be combined.

Response: Thanks for your good comments and suggestions. We have merged and rephrased these two WGD-related sections according to your suggestions to enhance the logical coherence of our manuscript (lines 154-168):

“To provide insights into the evolution history of the *R. pseudoacacia* genome, we conducted synonymous substitution rates (*Ks*) analyses on the genomes of *R. pseudoacacia*, *G. max*, and *M. truncatula*. This approach was utilized to identify the homologous gene pairs and to detect syntenic relationships between these species. The rate of *Ks* curves of collinear gene pairs suggested that the genomes of *R. pseudoacacia* underwent two rounds of whole-genome duplication (WGD). The older event, known as the γ event, is shared by all angiosperm lineages, while the more recent event occurred ~59 million years ago and is shared by all species in the order Leguminosae³³ (Fig. 2B). Notably, unlike *G. max*³⁴, *R. pseudoacacia* did not experience a species-specific WGD event (Fig. 2B). Furthermore, we detected a total of 88,836 gene pairs among these three species, and classified them into 2,304 syntenic blocks, covering 662 Mb (97.73%) of the *R. pseudoacacia* assembly. Consistent with the *Ks* analysis, we also found a 2:4 syntenic relationship between *R. pseudoacacia* and *G. max*, and a 2:2 syntenic relationship between *R. pseudoacacia* and *M. truncatula* (Figs. 2C and S4). A selected gene family was comprised of the same ratio of gene copies from different species analysed above, which also confirms these two WGD events (Fig. S5). These results provide additional evidence that there was no *R. pseudoacacia*-specific WGD event.”

2. The authors predicted a much high number of genes in *R. pseudoacacia* than in *C. arietinum* and *L. japonicus*, which they claimed to be a result of gene expansion. However I am curious whether this is the result of redundancy of haplotype sequences due to the assembly process?

Response: We thank the reviewer for raising this issue. We agree with the reviewer that redundancy of haplotype sequences can affect gene prediction and lead to overestimation of gene numbers. However, in our study, we took several measures to avoid such affections.

First, we used long-read sequencing data from Oxford Nanopore to assemble the genome. Long-read sequencing can provide more contiguous and complete assemblies, reducing the potential redundancy of haplotype sequences. Additionally, we performed polishing steps to refine the

assembly and minimize errors.

Second, we used a combination of *ab initio* gene prediction, transcriptome data, and protein homology information for genome annotation. This multi-evidence approach helps improve the accuracy of gene prediction and reduces the likelihood of including redundant gene models.

Third, we used the Purge Haplotigs software to identify and remove allelic contig. This step also ensured that the redundancy of haplotype sequences in our genome is controlled.

Fourth, we evaluated the completeness of our gene annotation using BUSCO analysis, which assesses the presence of conserved genes. This analysis recovers a Complete duplicated BUSCO score (D) of 3.16 %, which is comparable to other closely related species (Table S7), suggesting that the high number of genes annotated in the *R. pseudoacacia* genome was not caused by the redundancy of haplotype sequences.

The estimated genome size based on k-mer analysis was 693 Mb, and the final assembly size was 682 Mb, indicating that the assembly does not contain a significant amount of haplotypic redundancy. While there may still be some level of redundancy in haplotype sequences, it is unlikely to explain the substantial difference in gene numbers observed between *R. pseudoacacia* and the comparison species. Therefore, based on the comprehensive approach we took in genome assembly, annotation, and quality control. We are confident that the observed gene expansion in *R. pseudoacacia* is a genuine biological feature and not a result of haplotype redundancy.

3. The authors mentioned that *R. pseudoacacia* exhibited a comparable ratio of genomic repeats to closely related species, which supports the idea that the large size of the black locust genome may be driven by the explosion of repetitive elements. However, it should be pointedly pointed out in the article that although the genomic repeats ratio was comparable, the total length of repetitive sequences may be greater in *R. pseudoacacia*. Additionally, how the exact number was between *R. pseudoacacia*, *C. arietinum*, and *L. japonicus*, and whether the timing of the outbreak of *R. pseudoacacia* LTRs (since the authors also mention LTRs as the main type of repetitive sequences) could be added to the analysis to support this conclusion. This would provide the reader with a more convincing and clearer understanding of genomic repeats of *R. pseudoacacia*.

Response: Thank you for your insightful suggestions. Based on the comments, we provided more details on the total length of repetitive sequences and we have incorporated information about the timing of the outbreak of *R. pseudoacacia* LTRs to support our conclusion (Table S8, Fig. S3, lines 118-119; 126-129).

“Over 405.9 Mb of sequences were identified as repetitive elements. Together, they constituted 59.47% of the genome assembly that was predominately LTRs (Table S8).”

“The primary reason for this difference in genome size can be attributed to the influence of long terminal repeat retrotransposons (LTR-RTs) on genome size and evolution²⁹ (Fig. S3). LTR-RTs are responsible for over 75% of the genome in some plant species and have been identified as a significant driving force behind genome expansions³⁰.”

4. A statement of the threshold values should be added to the legend of Figure 5A/B, otherwise the reader can't clearly understand the figure notes of only the genes with significant values are shown.

Response: We have updated the legend of Figure 5A/B and also C-E to include the statement of the threshold values in the revised manuscript.

5. "Data acquired in this Whole Genome Shotgun project has been deposited in the BIG Data Center (<http://bigd.big.ac.cn>) under project number PRJCA011483."

I checked the BIG database but did not find any information related to the PRJCA011483 project in my search. If your manuscript is accepted, please make sure to publish all the raw data of sequencing.

Response: Thank you very much for your carefulness. We have now successfully uploaded all of raw data onto the National Genomics Data Center (<http://bigd.big.ac.cn>) for the PRJCA011483 project.

Minor Concerns:

1. Fig. 1B: Please use the same style in this table in a standardized way, either with or without parentheses.

Response: Thanks for your suggestion. We have standardized the style format of Fig. 1B based on your suggestion.

2. Legend of Fig 2C: Legend of Fig 2C: what is 4:2:2? Please add the detail. From the figure I can see it's *G. max* : *R. pseudoacacia* : *M. truncatula*. Please indicate in the legend the specific object of this scale.

Response: Thank you for your comments. We have revised the legend of Figure 2C accordingly. It now reads

"Collinear relationship between *R. pseudoacacia*, *G. max*, and *M. truncatula* is represented by a ratio of 4:2:2 (*G. max* : *R. pseudoacacia* : *M. truncatula*). Interspecific syntenic blocks that span more than 14,000 genes are indicated with lines, and some of the 4:2:2 blocks are highlighted in red".

3. Line 102-103: The authors mentioned the resulting assembly showed better continuity than other currently available genomes from the family Fabaceae, but I am not aware of any evidence to support this result. At least I can't find relevant evidence in Fig. 2A. It is suggested that N50 information of the other genomes be included in Fig. 2A to support this conclusion.

Response: Thank you for pointing out the confusion. The information supporting the claim of better continuity compared to other genomes from the family Fabaceae can be found in newly added

Supplementary Table S5, which includes the N50 values for various Fabaceae genomes. We have updated the manuscript accordingly to this comment (lines: 108-109).

4. Line 173-175: The term “closely related species” used here is ambiguous. How exactly is “closely related” defined in this context? Is it the extended gene set of *R. pseudoacacia* larger than that of all closely related species? Without evidence to support this, a clear description of *C. arietinum* and *L. japonicus* is suggested.

Response: Thank you for your good questions and valuable suggestions. This issue has been addressed in the text as:

“Our results showed 2,256 gene families were expanded in the *R. pseudoacacia* genome and ranked third among all the selected species (Fig. 2A).” at lines 176-177.

5. Line 213-216: The reader cannot get exactly which individuals are outliers from the supplementary table, and it is recommended to mark them in the Table S13.

Response: Thank you for your suggestion. We have marked the outlier individuals in Table S16 in our revised manuscript, as well as on the revised Figure 4 (A to C) for better clarity. This will make it easier for readers to identify the outlier individuals and their corresponding information.

6. Fig. 4C: The clustering of the different cities’ individuals cannot be obtained intuitively from the Fig. 4C, and the individuals that make up the trees are not visible from the trees, which makes it difficult to understand the clustering and grouping. At least for me, I am unable to find this information from Fig. 4C or the supplementary table.

Response: Thank you for your suggestions. We have updated Figure 4A and 4C to include specific individual information so that readers can more clearly understand the clustering and grouping information.

“Fig. 4. Population genomic analyses of *Robinia pseudoacacia*. (A) Geographic distribution of the *R. pseudoacacia* samples sequenced in this study. The population IDs were labeled in the figure. (B) Principal component analysis (PCA). The numbers in brackets represent the fraction of variance explained by each component. The outlier individuals were labeled in black text. (C) Maximum likelihood (ML) tree and ADMIXTURE analysis. The phylogenetic analysis was performed based on the whole-genome SNPs of 59 *R. pseudoacacia* individuals, with individual IDs marked alongside the corresponding sections of the ADMIXTURE results. For each ID, the prefix before “-” corresponds to the sampling site number displayed in panel (A), while the suffix after it indicates the individual number. Three *L. japonicus* individuals were used as the outgroup. (D) Demographic history inferred by the PSMC

model. (E) Observed genome-wide sequence diversity (π) for *R. pseudoacacia* and other species. The species names of cultivars (including landraces) and wild lineages were labelled in blue and black colours, respectively.”

7. Fig. 4E, Table S14: I could not get information from the Fig. 4E whether other species were cultivars or not. The description of the results here is overly simplistic and crude, and I did not even get valid information on cultivars and non-cultivars. Although the genetic diversity of *R. pseudoacacia* are relatively high in the results, the lack of information on cultivars and non-cultivars questions about the validity of the findings here.

Response: We have improved the Fig. 4E and primary Table S14 (named Table S17 in the current version of MS now), with the cultivars (including the landraces) labeled in blue color and the wild lineages in black color respectively, to show them more clearly.

Reviewer #2 (Remarks to the Author):

Wang et al., reported a whole genome sequence for *Robinia pseudoacacia* and re-sequenced the species across China. The paper is well organized and the quality of the genome is quite good. The weakness part is the population genomic analysis. The authors should describe the re-sequenced pops/individuals in detail. When and where all these individuals from? How its genotype distributed and genetic structure of the introduced pops? Do the genotype associate with the environment factors?

Response: Thank you for your feedback. We apologize for any confusion or lack of information in the previous version. We have made extensive revisions to address your concerns. In our revised manuscript, we provide a more detailed description of the re-sequenced populations and individuals of *Robinia pseudoacacia* across China. We include information on the locations for each population (Table S15), as well as details on the distribution of genotypes and the genetic structure of the introduced populations (Fig. S9).

Regarding the association between genotype and environmental factors, we have performed additional analyses to investigate this relationship. We have conducted tests for isolation by distance (IBD) and isolation by environment (IBE) to assess the genetic differentiation and potential associations with environmental factors. Our results (Fig. S6) suggest that there is no significant association between the genotype of *Robinia pseudoacacia* and the environmental factors, which is consistent with our original conclusion that *R. pseudoacacia* in these sampled counties were introduced through cultivation rather than natural dispersal.

Comments in detail:

L2: title: what insights? It is better to report the main conclusion directly in the title.

Response: Thank you for your suggestion. We have revised title to "Chromosome-level genome

assembly and population genomics of *Robinia pseudoacacia* reveal the genetic basis of its wide cultivation in urban greening". We think it more directly highlights the main conclusion of our study now.

L24: what comparative analysis?

Response: Thank you for the question. The comparative analyses conducted in this study mainly involved six parts:

1. Use of the software OrthoFinder to identify orthologous genes across different species.
2. Reconstruction of a phylogenetic tree using the identified orthologous genes.
3. Utilization of the PAML software to estimate the divergence time between the species.
4. Application of the CAFE software to identify gene families that have undergone expansion or contraction during the evolutionary history.
5. Focus on gene families that have expanded in black locust and investigation of their functional enrichment.
6. Identification of functional categories associated with traits preferred by urbanites, such as wood formation, biosynthesis, and drought tolerance.

These analyses allowed for a comprehensive comparative understand of the genomic data, providing insights into the evolutionary history and potential adaptations of black locust.

L42: "major considerations" is unclear here

Response: Thank you for pointing out this issue with our manuscript. You're correct in noting that the phrase "major considerations" in manuscript L42 was vague. What we intended to express here is that, due to rapid urbanization and the desire to satisfy the recreational demands and landscape perceptions of urbanites, the need for green infrastructure, particularly urban planting, has gained significant importance among Chinese city designers. We have changed the phrase here to

"The demand for green infrastructure, particularly in terms of urban planting, has become increasingly important for Chinese city designers due to rapid urbanization and the desire to meet the recreational needs and landscape perceptions of urbanites".

The content is in the revised manuscript of lines 42-45.

L53: describe the introduced history in detail and cite relevant references.

Response: Thank you for your good suggestion. In the revised manuscript, we have provided a more detailed description of the introduction history of *R. pseudoacacia* and included relevant references.

The revised sentence now reads as follows:

"*R. pseudoacacia* was first introduced to China and planted in the cities of Nanjing and Qingdao during the late nineteenth century¹⁷. " at lines 61-62.

L58-59: it is better not mix common and latin name.

Response: Thank you for your suggestions. In the revised version, we have substituted all instances of the common name "black locust" with the corresponding Latin name "*Robinia pseudoacacia*" thereby ensuring clarity and consistency in the text as suggested.

L62: Chinese cites, very vague description.

Response: Thank you for your feedback. We have revised the manuscript accordingly. The revised sentence in lines 65-68 now reads:

"Additionally, its high tolerance to a wide range of soil conditions and adaptability to many environmental stressors, such as temperature change, drought, air pollution, and low fertility enable it to thrive in Chinese urban areas across vast climatic regions while providing significant ecosystem services."

L133-139: To me, this paragraph is empty, without useful information, can be used in any plant species.

Response: We appreciate your feedback on this paragraph. After thoughtful consideration of your input, we agree that the paragraph is deficient in specific content. Therefore, we have decided to remove this paragraph from the text. We aim to provide valuable and relevant information, and your feedback assists us in enhancing the quality of our work.

L238-241: genetic diversity is not direct linked to fitness.

Response: Thank you for your good question. We agree that genetic diversity is not always directly linked to fitness. However, genetic diversity can exert a significant influence on populations' adaptive capacity to changing environments by affecting evolvability, that is, the capacity for further evolution (Barrick and Lenski 2013; Payne and Wagner 2019). Specifically, genetic diversity offers a broader range of traits and variations that can increase the likelihood of some individuals within a population having advantageous characteristics that enable them to better cope with environmental challenges. We have revised the section (lines 248-250) to better convey this idea:

"Genetic diversity can exert a significant influence on populations' adaptive capacity to changing environments by affecting evolvability, that is, the capacity for further evolution^{45,46}"

Barrick, J.E. and Lenski, R.E. (2013) Genome dynamics during experimental evolution. *Nature*

Reviews Genetics, 14, 827-839.

Payne, J.L. and Wagner, A. (2019) The causes of evolvability and their evolution. *Nature Reviews Genetics*, 20, 24-38.

L247: reorganize the sentence.

Response: Thank you for your suggestion. We have reorganized the sentence as follows (lines 259-261):

"To reveal the genetic basis of the traits favored by urbanites at the population level, we performed a comprehensive analysis of the whole genomes of 59 cultivated *R. pseudoacacia* trees planted in diverse environments and examined their genomic signatures for recent selective sweeps."

Reviewer #3 (Remarks to the Author):

Robinia pseudoacacia is an important street trees in the world, with high ecological and economic values. In this study, Wang et al. tried to explore its genetic characteristics for urban greening. Firstly, the authors reported a high-quality chromosome-level genome assembly of *R. pseudoacacia*, and demonstrated the genetic mechanism for its characteristic traits (especially wood formation) via comparative genomic analyses. Then, the population genomic analyses were carried out using 59 resequenced individuals collected from 14 counties. A total of 11 positively selected genes were further identified likely contributing to the urban greening-related traits. In summary, this research provides insights into the molecular mechanisms for urban greening, which is relatively novel in my opinion. The genetic resource provided in this research is also valuable for the future related studies. I think this MS is suitable for Communications Biology. However, there are still several issues that need to be addressed.

Response: Thank you for your positive comments on the manuscript. It's encouraging to hear that you find our research insightful and valuable for the field of urban greening. We also appreciate that you have raised some issues that help us to further refine our manuscript.

Major concerns:

1. The flower is a clear indication to recognize the *R. pseudoacacia*, which should be provided in Figure 1.

Response: Thank you for your good suggestion. We have revised Figure 1 to include an image of the flower of *Robinia pseudoacacia*.

2. The authors claimed LAC gene family was expanded in *R. pseudoacacia*, which may be involved in the process of wood formation. I wonder that whether this gene family suffers natural selection?

Whether they are identified as the positively selected genes? If not, why?

Response: Thanks very much for this good question. All the 31 genes from the LAC gene family in *R. pseudoacacia* were not identified as the positively selected genes (PSGs) in the subsequent analyses based on population genomic data. They may not suffer strong recent natural selection in *R. pseudoacacia*. Actually, we could speculate that, the PSGs are more likely to be conserved in the lineage for suffering strong natural selection. So, they always do not trend to expand. Furthermore, there are always many paralogs in the expanded gene family, which have the relatively high similarity between their sequences, leading to big challenges to examine whether they suffer the strong natural selection. Thus, there is no common agreement for the relationship between the expanded gene family and positively selected genes. We think the reviewer proposed a really good question, which we could deeply explore in future studies.

3. Gene families related to wood formation have been enriched in Figure 3A. These expanded families related to wood formation is important for a deeper understanding of black locust and should be provided in supplementary file.

Response: Thank you for your suggestions. The revised manuscript includes a detailed exhibit (the newly added Supplementary Table S13) that provides information on the enrichment of wood-related gene families and the genes they contain.

4. More details should be provided in the Methods section, such as the program parameters, so that we can better assess the accuracy of results and follow the methods of this research.

Response: Thank you for your valuable comments. In our revised manuscript, we have added more details regarding the software parameters in the Materials and Methods section.

5. L58-59: The Latin name of white clover should be provided.

Response: Thank you very much for pointing out the problems with our manuscript here. In the revised manuscript we have provided the Latin name of white clover (*Trifolium repens*) at lines 62-63.

6. L212: Please reconfirm whether PCs 1-10 could only explain a total of 14.15% genetic variance.

Response: Thank you for your comment regarding the explanation of genetic variance by PCs 1-10. We have reconfirmed the results, and it is accurate that PCs 1-10 can explain a total of 14.15% of the genetic variance. In order to provide further clarification and visualization, we have added a supplementary figure (Fig. S7) to the manuscript to illustrate the distribution of genetic variance explained by these principal components.

7. L353 and somewhere else: in “110x”, the letter “x” should be changed to a multiplication sign.

Response: Thank you for catching this. We have rechecked the manuscript carefully and changed the “x” in the manuscript to a multiplication sign “×”.

8. Figure 2C: The color of LACs in *R. pseudoacacia* and *A. thaliana* is too similar to distinguish.

Response: Thanks for bringing up this comment. We have updated Figure 2C to enhance the contrast between the color (and the shape of the lines) of LACs in *A. thaliana* and *R. pseudoacacia*, making it easier for readers to differentiate between the two.

9. Fig. 4A: The sampling location of individual sequenced for de novo genome should be labeled in the figure.

Response: Thank you for your suggestion. You are correct that we should mark the sampling location of the individual sequenced for de novo genome. We have revised Figure 4A to label the sample location in the figure 4A.

10. Fig. 4D: Please describe how to set the generation time (g) and mutation rate (μ) in the Methods section.

Response: Thanks for your comments. We have updated the details of using PSMC in the methods section with a description of the generation time (g) and mutation rate (μ) settings in the revised manuscript at lines 500-505.

REVIEWERS' COMMENTS:

Reviewer #1 (Remarks to the Author):

I have reviewed the revised manuscript, and found that the authors had largely revised it. So I recommend that the current version could be acceptable in its current form.

Reviewer #2 (Remarks to the Author):

I had finish review the revised ms. Please see editings in attach.
I would like to see the work be published as they revise the commetns.

Reviewer #3 (Remarks to the Author):

The suggestions I put forward previously have been well revised in the article, and the figures in the article have been updated and better displayed. This revised version is clearly better than the first version. I have no further questions about the content this time.

Reviewer #1 (Remarks to the Author):

I have reviewed the revised manuscript, and found that the authors had largely revised it. So I recommend that the current version could be acceptable in its current form.

Response: We are pleased to hear that the revisions have addressed your concerns and that the current version is acceptable to you. We appreciate your time and input throughout the review process.

Reviewer #2 (Remarks to the Author):

I had finish review the revised ms. Please see editings in attach.

I would like to see the work be published as they revise the comments.

Response: Thank you for your review of the revised manuscript. We appreciate the time and effort you have put into providing your comment and suggestions. We have carefully considered your comments and have made the necessary revisions to address them. We have also reviewed the attached edits and have incorporated them into the final version of the manuscript. We are pleased to hear that you are satisfied with the revisions and believe that the manuscript is now ready for publication.

Reviewer #3 (Remarks to the Author):

The suggestions I put forward previously have been well revised in the article, and the figures in the article have been updated and better displayed. This revised version is clearly better than the first version. I have no further questions about the content this time.

Response: Thank you for your positive comment regarding the revisions made to the article. We appreciate your thorough review and comments, and we are pleased that the revised version has met your expectations.